# Novel Insights for Patients with Multiple Basal Cell Carcinomas and Tumors at High-Risk for Recurrence: Risk Factors, Clinical Morphology, and Dermatoscopy

**DOI:** 10.3390/cancers13133208

**Published:** 2021-06-27

**Authors:** Dimitrios Sgouros, Dimitrios Rigopoulos, Ioannis Panayiotides, Zoe Apalla, Dimitrios K. Arvanitis, Melpomeni Theofili, Sofia Theotokoglou, Anna Syrmali, Konstantinos Theodoropoulos, Georgia Pappa, Vasileia Damaskou, Alexander Stratigos, Alexander Katoulis

**Affiliations:** 12nd Department of Dermatology-Venereology, “Attikon” General University Hospital, Medical School, National and Kapodistrian University of Athens, 12462 Athens, Greece; arvandi18@yahoo.com (D.K.A.); melpomenitheofili@hotmail.com (M.T.); theotokoglousofia@gmail.com (S.T.); annasyrmali@gmail.com (A.S.); theod28@gmail.com (K.T.); mdgeopap@med.uoa.gr (G.P.); akatoulis@med.uoa.gr (A.K.); 21st Department of Dermatology-Venereology, Andreas Sygros Hospital, Medical School, National and Kapodistrian University of Athens, 16121 Athens, Greece; drigopoul@med.uoa.gr (D.R.); alstrat@med.uoa.gr (A.S.); 32nd Department of Pathology, “Attikon” General University Hospital, Medical School, National and Kapodistrian University of Athens, 12462 Athens, Greece; ioagpan@med.uoa.gr (I.P.); damaskou@0320.syzefxis.gov.gr (V.D.); 4State Clinic of Dermatology, Hospital for Skin and Venereal Diseases, 54643 Thessaloniki, Greece; zapalla@auth.gr

**Keywords:** basal cell carcinoma, dermatoscopy, histopathology, skin cancer, diagnosis, non-melanoma skin cancer, prevention

## Abstract

**Simple Summary:**

We investigated 225 patients with 304 primary basal cell carcinomas (BCCs) and we conducted a retrospective, morphological, cohort study aimed at evaluating patients’ demographics and tumors’ clinical and dermatoscopic characteristics. Our main objectives were the detection of risk factors for multiple BCCs in individual patients and the description of clinical and dermatoscopic features of low and high risk for local recurrence tumors. The rising incidence of BCC and the occurrence of multiple tumors in individual patients poses BCC as a major issue for health systems. To the best of our knowledge, this is one of the first studies to attempt to unveil clinical and dermatoscopic features of low-/high-risk neoplasms beyond histopathology and take into equal account parameters, such as anatomic location and size of the lesion. We strongly support that profiling of multiple patients with BCCs and a thorough knowledge of high-risk tumors’ clinico-dermatoscopic morphology could provide physicians with important information towards prevention of this neoplasm.

**Abstract:**

Introduction: Basal cell carcinoma (BCC) quite frequently presents as multiple tumors in individual patients. Neoplasm’s risk factors for local recurrence have a critical impact on therapeutic management. Objective: To detect risk factors for multiple BCCs (mBCC) in individual patients and to describe clinical and dermatoscopic features of low- and high-risk tumors. Materials & Methods: Our study included 225 patients with 304 surgically excised primary BCCs. All patients’ medical history and demographics were recorded. Clinical and dermatoscopic images of BCCs were evaluated for predefined criteria and statistical analyses were performed. Results: Grade II-III sunburns before adulthood (OR 2.146, *p* = 0.031) and a personal history of BCC (OR 3.403, *p* < 0.001) were the major predisposing factors for mBCC. Clinically obvious white color (OR 3.168, *p* < 0.001) and dermatoscopic detection of white shiny lines (OR 2.085, *p* = 0.025) represented strongly prognostic variables of high-risk BCC. Similarly, extensive clinico-dermatoscopic ulceration (up to 9.2-fold) and nodular morphology (3.6-fold) raise the possibility for high-risk BCC. On the contrary, dermatoscopic evidence of blue-black coloration had a negative prognostic value for high-risk neoplasms (light OR 0.269, *p* < 0.001/partial OR 0.198, *p* = 0.001). Conclusions: Profiling of mBCC patients and a thorough knowledge of high-risk tumors’ clinico-dermatoscopic morphology could provide physicians with important information towards prevention of this neoplasm.

## 1. Introduction

Basal cell carcinoma (BCC) represents the most common type of skin cancer and human malignancy overall [1]. Demographics, etiopathogenesis, risk factors, histopathology, and clinico-dermatoscopic presentation of this tumor are well-described in the current literature [2,3,4]. Despite its extremely rare metastatic potential, the rising incidence of this neoplasm poses a major health issue for patients and healthcare systems [5]. Surgical excision is the treatment of choice for skin cancers. However, there are several efficient, guideline-approved, non-surgical, therapeutic options for specific subtypes of BCC (e.g., superficial) [2,3,6]. Current evidence shows that the classification of BCC as low or high risk for local recurrence, based on several clinical and histopathological characteristics (e.g., lesion’s maximum diameter, anatomic location, histologic subtype, etc.) can be decisive for the selection of treatment in everyday clinical practice [4,6].

Moreover, BCC quite frequently presents as multiple (>1) tumors in individual patients. In addition, a personal history of at least one BCC yields a 17-fold risk for a subsequent BCC [5]. Multiple BCC patient risk factors (e.g., sex, age, etc.) have been investigated in the literature [7,8,9,10]. Further studies on the clinical features of multiple tumors and patients’ profiling could provide physicians with important knowledge towards prevention of this neoplasm.

Dermatoscopic examination is a non-invasive, safe, and patient-friendly procedure that enhances a physician’s diagnostic accuracy. The use of this method for diagnosing BCC has been thoroughly investigated [11,12,13,14]. In addition to its diagnostic role, dermatoscopy can help as a follow-up measure for size reduction of locally advanced BCC under neo-adjuvant systemic treatments [15]. Of note is the correlation of currently accepted dermatoscopic criteria with histological subtypes of BCC [16,17,18,19].

Our study had a double primary objective, i.e., to detect risk factors for multiple BCC in individual patients and to describe clinical and dermatoscopic features of low and high-risk tumors in a population of 225 patients with 304 primary BCCs. As a secondary goal, we investigated the dermatoscopic findings in a subgroup analysis among solitary BCC and multiple BCCs, as well as high-risk tumors with histologically aggressive subtypes.

## 2. Materials and Methods

This was a retrospective, morphological, cohort study conducted by the 2nd Dermatology Department of the “ATTIKON” University Hospital of Athens (waiver decision by Ethics Committee 1248/19-1-2016). Informed consent was obtained from all subjects involved in the study.

The inclusion criterion was patients with a histopathological diagnosis of primary BCC that was surgically excised. The exclusion criteria were: (1) Gorlin–Goltz syndrome patients, (2) severely immunocompromised patients (i.e., patients under immunosuppressive treatment for autoimmune diseases or internal malignancies and HIV patients), (3) patients with locally recurrent tumors from prior treatments, and (4) patients that did not give their consent for data collection for the purposes of the study. The enrollment period was between January 2016 and January 2018. All selected patients had a thorough physical skin examination and a full report of their medical history. In addition, a detailed history of sun exposure habits, previous cutaneous diseases, and past treatments for skin malignancies were recorded. Clinical and dermatoscopic images of the suspicious lesions were both captured at the initial medical visit before surgical excision using a Nikon J1 camera (Tokyo, Japan) and a handheld Dermlite Hybrid II dermatoscope (3Gen Inc, San Juan Capistrano, CA, USA). All the clinical and dermatoscopic characteristics of tumors were retrospectively evaluated for predefined criteria (Appendix A) [20] by two investigators non-blinded to the final diagnosis (D.S. and A.K.). The cohort was divided into two groups of patients, i.e., those with solitary neoplasms and those with more than one synchronous tumors (solitary vs. multiple BCC). The tumors were classified into low- and high-risk tumors for recurrence based on three factors: (1) tumor size, (2) anatomic location, and (3) histological subtype (Table 1) [6]. Of note, we did not include the “clinical margins criterion” as a risk factor for tumor classification, since we planned to investigate it as an independent clinical feature. Statistical analysis with uni- and multivariate logistic regression was performed for demographic factors along with clinical and dermatoscopic features for the above-mentioned groups of patients and tumors.

### Statistical Analysis Methods

The Shapiro–Wilk and Shapiro–Francia tests were used for normality of distribution. Continuous variables following a normal distribution are presented as a mean ± standard deviation, whereas not normally distributed variables are presented as a median with interquartile range (25th and 75th percentiles). For categorical variables, the frequencies and percentages were used. Chi-squared and Fischer’s exact tests were used for the comparison of categorical variables, while unpaired t-tests and Mann–Whitney U tests were applied depending on the distributions of the continuous variables. Univariate and multivariate logistic regression was also performed. All statistical calculations were based on a two-sided hypothesis, and a *p*-value of <0.05 was considered to be statistically significant. All statistical analyses were performed using Stata/IC version 15.1 (StataCorp, Lakeway Drive, Texas, USA).

## 3. Results

### 3.1. Solitary (sBCC) and Multiple BCCs (mBCC): Patients’ Demographics

In total, 225 patients with 304 primary BCCs were included. There were 172 patients (76.4%) who presented with a solitary tumor and 53/225 patients (23.6%) were diagnosed with ≥2 tumors at the initial evaluation visit. The male sex prevailed in both groups of patients with 105/172 (61.1%) and 38/53 (71.2%) for sBCC and mBCC, respectively. The median age for the entire group of patients was 73 years. However, patients with mBCC (median age 75) were older than sBCC patients (median age 72.5). Skin exposure habits and chronic solar damage were more prominent in the group of patients with mBCC rather than the solitary neoplasms group. In specific, 24/53 patients (45.3%) reported occupational sun exposure, 19/53 patients (35.9%) had a history of at least one severe sunburn (≥grade II) during childhood-adolescence and 33/53 patients (62.3%) were diagnosed with actinic keratoses as compared with 58/172 (33.7%), 38/172 (22.1%) and 84/172 (48.8%) patients, respectively, for the sBCC group. A personal history of any type of skin cancer and history of at least one previous BCC were two independent risk factors, more prevalent in the group of multiple tumors (24/53 (45.3%) and 23/53 (43.4%) patients, respectively) as compared with the sBCC group (41/172 (23.8%) and 33/172 (19.2%) patients). All the aforementioned results can be seen in Table 2.

Univariate logistic regression showed that a personal history of BCC (3.2-fold), a personal history of skin cancer (2.6-fold), sunburns grade II–III (<18 years old) (1.9-fold), and the presence of actinic keratosis (1.7-fold) were important risk factors for mBCC. However, the multivariate analysis revealed that severe sunburns during childhood-adolescence and a personal history of BCC were the two most critical risk factors for the development of ≥2 BCC in an individual patient with a 2.1-fold and 3.4-fold risk, respectively (Table 3).

### 3.2. Solitary BCC and Multiple BCCs: Tumors’ Clinical and Histological Features

In total, 172 patients presented with one BCC and 53 patients had multiple lesions (132 overall). Specifically, 38 patients presented with two tumors (38/53, 71.7%) nine patients with three tumors (9/53, 17.1%), three patients with four tumors (3/53, 5.6%), and ≥5 tumors were detected in three patients (3/53, 5.6%). Regarding important factors of risk stratification for local recurrence (such as diameter, anatomic site, and histopathology) no striking difference was recorded between the two groups of patients. Specifically, sBCC comprised of 46/172 patients (26.7%) with low-risk tumors and 126/172 patients (73.3%) with high-risk tumors, while mBCC consisted of 46/132 patients (34.9%) with low-risk tumors and 86/132 patients (65.2%) with high-risk neoplasms. In both groups, the head/neck area was the most common anatomic location for the development of BCC (73.4%), followed by the trunk (19.1%) and extremities 7.6%). The median diameter of the neoplasms was 0.9 cm. Regarding histologic subtypes of BCC, our results showed a prevalence of types of indolent biologic behavior (i.e., nodular and superficial, 73.7%) rather than aggressive growth pattern forms (i.e., infiltrative, morpheaform, basosquamous, micro-nodular, and mixed, 26.3%). Specific results for the subgroups of sBCC and mBCC are shown in Appendix A.

### 3.3. Low Risk versus High Risk for Local Recurrence BCC: Clinical Characteristics

Out of the 304 tumors that were included in the study, 92 tumors were classified as low-risk tumors and 212 tumors were classified as high-risk tumors based on three criteria, i.e., lesion’s maximum diameter, anatomic location, and histopathology (Table 1). Clinical margins were investigated as a separate feature. Indeed, well-defined clinical borders were predominant (78/92, 84.8%) among the subgroup of low-risk BCC, as expected, although poorly defined clinical margins were not a prevalent characteristic in high-risk tumors either (82/212, 38.7%). In terms of ulceration, intact epidermis was evident in the majority of low-risk tumors (49/92, 53.3%), in contrast to high-risk tumors which exhibited prominent erosion/ulceration (155/212, 73.1%). Clinically, most of the high-risk tumors presented as nodular lesions (146/212, 68.9%) while 50% (46/92) of low-risk BCC had a nodular morphology. It is worthwhile mentioning that an important subset of low-risk tumors was flat (23/92, 25%) as compared with 5.7% (12/212) in high-risk BCC. Concerning coloration, pink was the most frequently observed color in both subgroups with a total proportion of 78.6% (239/304). A white color was more evident among high-risk tumors (54.7% versus 28.3%), while a blue-black color was more commonly encountered among low-risk BCC (51.1% versus 31.13%) (Table 4).

Univariate logistic regression for the clinical features of high-risk versus low-risk BCC is presented in Appendix A. Multivariate logistic regression showed that extensive clinical ulceration (>90% of total lesion surface) and prominent ulceration yield a 9.2-fold and 2.5-fold probability for high-risk BCC, respectively. In the same context, white color and nodular morphology are strong prognostic factors for a high-risk tumor with an OR 3.6. On the contrary, clinical blue-black coloration is a negative prognostic factor for high-risk neoplasms (OR 0.2) (Table 4). A detailed analysis of multivariate logistic regression is presented in Appendix A.

### 3.4. Low Risk versus High Risk for Local Recurrence BCC: Dermatoscopic Features

Vascular structures were the most striking dermatoscopic finding in both subgroups of low-risk and high-risk tumors (272/304, 89.5%). Arborizing vessels prevailed in both subgroups as well (247/304, 81.2%) followed by telangiectasias and glomerular vessels. Telangiectasias were more common among low-risk tumors (38% versus 22.2%) and glomerular vessels were more frequent in high-risk tumors (14.2% versus 4.4%). As expected, high-risk tumors were mostly eroded or ulcerated lesions (168/212, 79.3%), while dermatoscopic erosion/ulceration was also prevalent in low-risk tumors (57/92, 62%). In terms of pigmentation, the majority of high-risk tumors were non-pigmented (115/212, 54.3%), while 71.4% (66/92) of low-risk BCC had dermatoscopic signs of pigmentation. In specific, all types of pigmented structures were more frequently observed in low-risk tumors as compared with high-risk tumors. Finally, dermatoscopic clues for white coloration were more frequently observed among high-risk tumors; white shiny lines (46.2% versus 27.2%); multiple yellow-white globules (12.7% versus 7.6%); white circles and yellow clods (25.9% versus 7.6%) (Table 5).

Univariate logistic regression for the dermatoscopic findings of high-risk versus low-risk BCC is presented in Appendix A. The multivariate analysis revealed that extensive (8-fold) as well as prominent (2.4-fold) ulceration, glomerular vessels (3.3-fold), and white shiny linear structures (2-fold) are positive predictive factors for a high-risk BCC. On the contrary, pink-whitish background (0.37-fold) along with pigmentation of any extent (0.2–0.3-fold) represent negative prognostic factors for high-risk tumors (Table 5 and Appendix A and Figure 1 and Figure 2).

### 3.5. Dermatoscopic Features in Subgroup Analysis for Solitary BCC and Multiple BCCs and Aggressive Histologic Subtypes

Dermatoscopic features of BCC in the subgroups of solitary and multiple tumors can be seen in detail in Appendix A. No striking differences were detected in the frequencies of dermatoscopic findings between both subgroups of patients, except for “white features”. In specific white shiny lines (51.7% versus 25.8%), multiple yellow-white globules (16.3% versus 4.6%) and white circles and yellow clods (24.41% versus 14.4%) were more commonly observed in sBCC rather than in the group of multiple tumors.

Appendix A shows in detail the dermatoscopic features in aggressive histologic subtypes of BCC as compared with the group of high-risk tumors. The specific subtypes can also be seen in Appendix A. Of note, there are no significant differences in the dermatoscopic findings of histologically aggressive tumors as compared with high-risk BCC apart from a slight prevalence of “white structures” already observed in the subgroup analysis for solitary tumors (Appendix A).

## 4. Discussion

### 4.1. Solitary versus Multiple BCCs

BCC morbidity represents a major issue for public health. Despite its extremely low mortality, the rising incidence of the tumor and the high occurrence of mBCC quantify the burden of disease comparable with esophageal, ovarian, or thyroid cancer, according to the WHO [21]. Profiling of patients with multiple tumors, identification of risk factors, and particular clinico-dermatoscopic features of mBCC could be valuable for the overall management of the disease.

Our study showed that grade II–III sunburns before adulthood (OR 2.146, *p* = 0.031) and a personal history of BCC (OR 3.403, *p* < 0.001) were the major predisposing factors for mBCC (Table 3). So far, the personal history of a previously treated BCC is considered to be a well-established risk factor for a subsequent BCC [10,22,23]. Chronic UV exposure has also been proven to be an important contributor for non-melanoma skin cancer [5]. However, there was no clear association of the sun exposure pattern with a single BCC or multiple BCCs [22]. According to our results, at least one severe sunburn during childhood or adolescence increases the risk for the development of mBCC later in life. Thus, preventive measures in early life could have a protective role against mBCC.

In our study, we observed that older patients at a slightly higher male/female ratio comprised the group of mBCC as compared with the sBCC group (Table 2). This finding was in accordance with the previously published literature [8,9,22]. The distribution of histological subtypes of the tumor in both groups was similar and corresponded to the average of the incidence of various forms of the neoplasm [1] (Appendix A). This result does not confirm previously published data that superficial BCC is a more common subtype in patients with multiple tumors [7,8,9,10].

We also performed a subgroup analysis for the dermatoscopic features in sBCC and mBCC. (Appendix A) No clear differences were detected between both groups, apart from a more frequent representation of white dermatoscopic structures (i.e., white shiny lines, white peri-follicular circles, multiple yellow and white globules) in the group with single BCC. White coloration in dermatoscopy of BCC is strongly correlated with collagen alteration, calcification, and thus deeper infiltration in dermis [24,25,26,27]. Our observation supports that the presence of multiple tumors is not necessarily associated with more aggressive BCC subtypes.

### 4.2. High-Risk versus Low-Risk BCC

Due to the extremely rare metastatic potential of BCC, the traditional classification for neoplasms is not applicable in this type of skin cancer. Thus, the tumors are categorized accordingly to the risk for local recurrence [3,6]. Our study evaluated BCCs as low/high risk based on three criteria (i.e., lesion’s diameter, anatomic location, and histology) in order to investigate predefined criteria for risk stratification as independent variables (i.e., clinical margins). Moreover, the exclusion of other factors (i.e., locally recurrent tumor, mBCC syndromes, and severe immunosuppression) allowed us to detect characteristics of primary lesions without the statistical bias that may arise from patients’ health status or physicians’ previous topical treatments (Table 1).

Of note, 69.7% (212/304) of tumors were staged as high risk, although only 26.3% (80/304) of tumors had a histologic subtype with aggressive behavior. This finding highlights the importance of the size and the site of the lesion, as equally significant along with histopathology, for risk stratification, and therefore treatment selection (Table 4 and Appendix A).

In terms of clinical morphology, low-risk BCC had mostly well-defined clinical margins (84.8%, 78/92), as expected. Surprisingly, high-risk tumors did not present with ill-defined borders as a prevalent feature (38.7%, 82/212) (Table 4). This result could be suggestive of the fact that a well-defined BCC is not always a low-risk tumor, and therefore other risk factors should be considered. Regarding clinically or dermatoscopically prominent ulceration and nodular presentation, we confirmed current evidence that both features typically characterize more aggressive subtypes of BCC [4,17,18,19,28,29,30] (Table 4, Table 5, Appendix A).

Coloration represents quite an interesting finding both clinically and dermatoscopically. Our results present clinically obvious white color (OR 3.168, *p* < 0.001) and dermatoscopically detected white shiny lines (OR 2.085, *p* = 0.025) as variables strongly prognostic of high-risk BCC. The latter is in accordance with the pre-existing literature for white dermatoscopic structures as features correlating with aggressive subtypes of BCC [17,18,19,24,25,26,27,28,29,30]. On the contrary, pigmentation due to melanin elements seems to be inversely associated with high-risk tumors. In terms of clinically apparent blue-black hue, the multivariate analysis showed a tendency for low-risk BCC (OR 0.193, *p* = 0.071). However, dermatoscopic evidence of blue-black coloration had a negative prognostic value for high-risk neoplasms (light OR 0.269, *p* < 0.001/partial OR 0.198, *p* = 0.001). This finding suggests that melanin may represent a positive predictive factor for BCC with a higher risk for local recurrence and supports recently published data on the hypothesis that well-differentiated non-aggressive BCC could preserve relatively more melanocytes [31] (Table 4, Table 5, Appendix A).

Concerning other dermatoscopic observations, we confirmed that arborizing vessels are typical for BCC and telangiectasias are mostly seen in low-risk tumors (Table 5). Of note, glomerular vessels in multivariate analysis were prognostic for high-risk BCC (OR 3.314, *p* = 0.044), another finding in line with the current literature [28,29]. Dermatoscopic evidence of ulceration also raised the possibility of a high-risk tumor (up to 8-fold) in accordance with the pre-existing published data [17,18,19,20,28,29,30] (Table 5).

Finally, we performed a subgroup comparative analysis investigating the occurrence of dermatoscopic variables between high-risk tumors (*n* = 212) and histologic subtypes of BCC with a more aggressive growth pattern (*n* = 80). The results are listed in Appendix A and surprisingly we could not detect any statistically important variations among the two groups. Our observations support the significance of the current classification system for BCC in clinical practice and show that factors other than histopathology (i.e., tumor’s diameter and anatomic location) have a critical impact on risk stratification, almost equal to histologic subtype. Thus, from a clinician’s perspective, clinical morphology and dermatoscopic findings could efficiently negate unnecessary biopsies and prevent therapeutic pitfalls.

### 4.3. Limitations

Our study has certain limitations including the “non-blinded to diagnosis” investigators evaluating clinical and dermatoscopic findings, the lack of a control group, and certain personal history information that were unavailable (e.g., age of first BCC diagnosis) and are considered by literature significant risk factors for mBCC. Moreover, the current classification system of BCC does not take into account the different impacts of each risk factor and this is an additional limitation in our study. Finally, the visit-seeking attitude of patients with a personal history of BCC and the surveillance plan after diagnosis of an initial tumor might pose a bias.

## 5. Conclusions

In conclusion, a personal history of BCC and moderate-to-severe sunburns during childhood-adolescence are the two most important risk factors for the development of multiple BCC. The presence of multiple tumors does not seem to be related to more aggressive subtypes of the neoplasm. Concerning high-risk for recurrence BCC, ill-defined clinical margins is not an absolute criterion. Nodular morphology, clinical and dermatoscopic evidence of ulceration, and the color white, either clinically obvious or presented dermatoscopically in the form of white shiny lines, serve as strong predictors for high-risk BCC. On the contrary, pigmentation due to melanin represents a negative prognostic value for high-risk tumors. To the best of our knowledge, this is one of the first attempts to describe the dermatoscopic features of locally aggressive BCC beyond the histologic subtype. We strongly support the need for further studies to unveil important diagnostic clues for BCC.

## Figures and Tables

**Figure 1 cancers-13-03208-f001:**
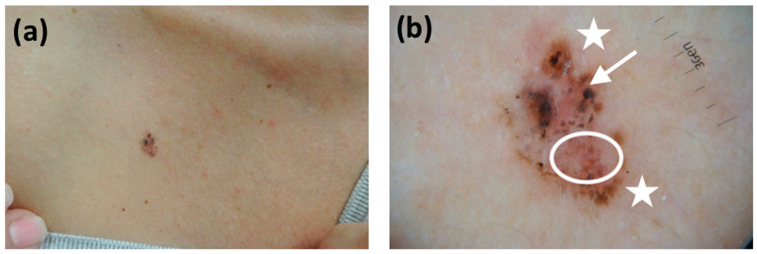
(**a**) A brown-black plaque on the chest of a 51-year-old female patient. The lesion has a maximum diameter of 0.6 cm. Histology set the diagnosis of a superficial BCC; (**b**) dermatoscopy confirmed our observations for low-risk neoplasms. Pigmentation was the striking feature in this tumor with leaf-like structures at the periphery (white asterisks), concentric structures (white circle), and a hint of telangiectasias (white arrow).

**Figure 2 cancers-13-03208-f002:**
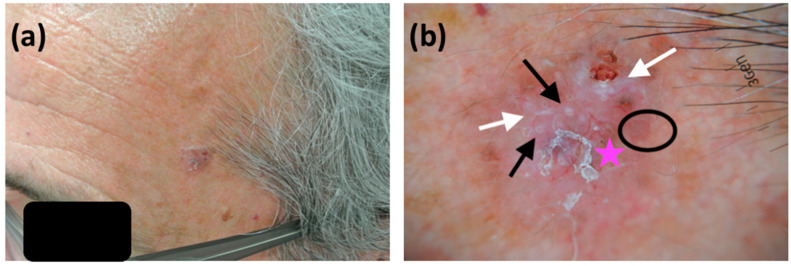
(**a**) A pinkish nodule on the left temple of a 66-year-old male patient with dark skin phototype and a maximum diameter of 1.3 cm, histopathologically diagnosed as a mixed BCC (nodular and metatypical). Due to anatomic location and size, the lesion was treated as a high-risk tumor for local recurrence; (**b**) dermatoscopic evaluation was in line with histology. A combination of arborizing (pink asterisk) and hairpin (black circle) vessels was evident. White, shiny linear structures (white arrows) were also obvious on the lesion’s surface. Finally, white, perifollicular circles along with central yellow clods were dermatoscopically apparent (black arrows).

**Table 1 cancers-13-03208-t001:** Risk factors for low and high risk for recurrence BCC ^1^.

Risk Factors	Low-Risk BCC	High-Risk BCC
**Location/size**	Trunk, extremities < 2 cm	Trunk, extremities ≥2 cm Cheeks, forehead, scalp, neck and pretibial any size “Mask areas” of face ^2^, genitalia, hands and feet
**Histopathology**	Nodular, superficial	Aggressive growth pattern ^3^

^1^ Any risk factor places the patient in the high-risk category; ^2^ center of face, eyelids, eyebrows, periorbital, nose, lips (cutaneous and vermilion), chin, mandible, pre- and post-auricular skin/sulci, temple, and ear; ^3^ infiltrative, basosquamous, morpheaform, micronodular, mixed, sclerosing/carcinosarcomatous features, perineural invasion. This table was adapted by Schmultz, C., Blitzblau, R., et al. Basal Cell Skin Cancer Version 2.2021 in NCCN Clinical Practice Guidelines in Oncology, available online at https://www.nccn.org/professionals/physician_gls/pdf/nmsc.pdf (accessed on 25 February 2021).

**Table 2 cancers-13-03208-t002:** Patients’ demographics with solitary BCC and multiple BCCs (*n* = 225).

	Total (*n* = 225)	Solitary BCC (*n* = 172)	Multiple BCCs (*n* = 53)
**Age**, median years (range)	73 (28–94)	72.5 (28–91)	75 (37–94)
**Sex,** ***n*** **(%)**			
Males	143 (63.6)	105 (61)	38 (71.7)
Females	82 (36.4)	67 (39)	15 (28.3)
**Fitzpatrick skin phototype, ** ***n*** **(%)**			
I	0	0	0
II	47 (20.9)	34 (19.8)	13 (24.5)
III	126 (56)	96 (55.8)	30 (56.6)
IV	52 (23.1)	42 (24.4)	10 (18.9)
**Occupational sun exposure, ** ***n*** **(%)**	82 (36.4)	58 (33.7)	24 (45.2)
**History of sunburns, ** ***n*** **(%) ***	57 (25.3)	38 (22.1)	19 (35.9)
**Actinic keratosis, ** ***n*** **(%)**	117 (52)	84 (48.8)	33 (62.3)
**Personal history of skin cancer, ** ***n*** **(%)**	65 (28.9)	41 (23.8)	24 (45.3)
**Family history of skin cancer, ** ***n*** **(%)**	24 (10.7)	19 (11.1)	5 (9.4)
**Personal history of BCC,** ***n*** **(%)**	56 (24.9)	33 (19.2)	23 (43.4)

* Grade II/III sunburns < 18 years old.

**Table 3 cancers-13-03208-t003:** Uni- and multivariate logistic regression for multiple BCCs vs. solitary BCC.

Univariate	*p*-Value	OR	95% CIs
Age	0.453		
Sex	0.159		
Fitzpatrick skin phototype	0.611		
Occupational Sun Exposure	0.126		
Personal history of skin cancer	0.003	2.644	1.388–5.038
Family history of skin cancer	0.74		
Personal history of BCC	0.001	3.229	1.665–6.265
Actinic keratosis	0.089	1.729	0.92–3.248
History of sunburns	0.046	1.971	1.011–3.84
**Multivariate**	***p*** **-Value**	**OR**	**95% CIs**
Personal history of BCC	<0.001	3.403	1.732–6.685
History of sunburns	0.031	2.146	1.073–4.295

For the final model, a fitness of good control was performed based on Hosmer–Lemeshow criterion (*p*-value = 0.648).

**Table 4 cancers-13-03208-t004:** Low-risk and high-risk tumors’ clinical characteristics and multivariate analysis.

	**Total (*n* = 304)**	**Low-Risk (*n* = 92)**	**High-Risk (*n* = 212)**	**Multivariate** **OR (95% CI/*p-*Value)**
**Margins,** ***n*** **(%)**				
Well-defined	208 (68.4)	78 (84.8)	130 (61.3)	
Ill-defined	96 (31.6)	14 (15.2)	82 (38.7)	2.007 (0.952–4.23/0.067)
**Ulceration,** ***n*** **(%)**				
None	106 (34.9)	49 (53.3)	57 (26.9)	
Erosions	55 (18.1)	21 (22.8)	34 (16.)	
Prominent	108 (35.5)	20 (21.7)	88 (41.5)	2.533 (1.243–5.162/0.011)
>90%	35 (11.5)	2 (2.2)	33 (15.6)	9.241 (1.79–47.711/0.008)
**Clinical presentation, ** ***n*** **(%)**				
Flat	35 (11.5)	23 (25)	12 (5.6)	
Elevated	77 (25.3)	23 (25)	54 (25.5)	2.384 (0.892–6.376/0.083)
Nodular	192 (63.2)	46 (50)	146 (68.9)	3.674 (1.502–8.988/0.004)
**Colors,** ***n*** **(%)**				
Pink color	239 (78.6)	75 (81.5)	164 (77.4)	
White color	142 (46.7)	26 (28.3)	116 (54.7)	3.682 (1.988–6.819/<0.001)
Blue-black color	113 (37.2)	47 (51.1)	66 (31.1)	0.193 (0.032–1.153/0.071)
**Pigmentation intensity, ** ***n*** **(%)**				
None	183 (60.2)	44 (47.8)	139 (65.6)	
Light	44 (14.5)	20 (21.7)	24 (11.3)	
Partial	36 (11.8)	15 (16.3)	21 (9.9)	
Heavy	41 (13.5)	13 (14.2)	28 (13.2)	5.611 (0.771–40.82/0.088)

For the final model, a fitness of good control was performed based on Hosmer–Lemeshow criterion (*p*-value = 0.547).

**Table 5 cancers-13-03208-t005:** Low-risk and high-risk tumors’ dermatoscopic features and multivariate analysis.

	Total (*n* = 304)	Low-Risk (*n* = 92)	High-Risk (*n* = 212)	Multivariate OR (95% CI/*p-*Value)
**Vasculature,** ***n*** **(%)**				
None	32 (10.5)	13 (14.1)	19 (9)	
Apparent (<50%)	219 (72.1)	63 (68.5)	156 (73.6)	
Prominent (≥50%)	53 (17.4)	16 (17.4)	37 (17.4)	
**Vessels,** ***n*** **(%)**				
Arborizing	247 (81.3)	65 (70.7)	182 (85.9)	
Telangiectasias	82 (27)	35 (38.)	47 (22.2)	
Glomerular	34 (11.2)	4 (4.4)	30 (14.2)	3.314 (1.033–10.626/0.044)
Linear irregular	14 (4.6)	2 (2.2)	12 (5.7)	
Dotted	1 (0.3)	0	1 (0.5)	
Hairpin	3 (1)	0	3 (1.4)	
Polymorphous	14 (4.6)	0	14 (6.6)	
**Pigmented structures, ** ***n*** **(%)**				
Blue-gray ovoid globules	127 (41.8)	46 (50)	81 (38.2)	
Multiple dots	90 (29.6)	40 (43.5)	50 (23.6)	
Spoke-wheel	24 (7.9)	15 (16.3)	9 (4.3)	
Leaf-like	28 (9.2)	18 (19.6)	10 (4.7)	
Concentric	16 (5.3)	10 (10.9)	6 (2.8)	
**Pigmentation intensity, ** ***n*** **(%)**				
None	141 (46.4)	26 (28.2)	115 (54.2)	
Light (<10%)	77 (25.3)	33 (35.9)	44 (20.8)	0.269 (0.13–0.558/<0.001)
Partial (10–50%)	42 (13.8)	18 (19.6)	24 (11.3)	0.198 (0.078–0.5/0.001)
Heavy (>50%)	44 (14.5)	15 (16.3)	29 (13.7)	0.313 (0.105–0.934/0.037)
**Pink-whitish background, ** ***n*** **(%)**	211 (69.4)	71 (77.2)	140 (66)	0.369 (0.158–0.862/0.021)
**Diffuse white color, ** ***n*** **(%)**	11 (3.6)	1 (1.1)	10 (4.7)	
**White shiny lines, ** ***n*** **(%)**	123 (40.5)	25 (27.2)	98 (46.2)	2.087 (1.097–3.971/0.025)
**Multiple yellow-white globules, ** ***n*** **(%)**	34 (11.2)	7 (7.6)	27 (12.7)	
**White circles and yellow clods, ** ***n*** **(%)**	62 (20.4)	7 (7.6)	55 (25.9)	
**Ulceration,** ***n*** **(%)**				
None	79 (26)	35 (38)	44 (20.8)	
Erosions	71 (23.3)	30 (32.6)	41 (19.3)	
Prominent	121 (39.8)	25 (27.2)	96 (45.3)	2.451 (1.198–5.014/0.014)
>90%	33 (10.9)	2 (2.2)	31 (14.6)	8.042 (1.637–39.505/0.01)

For the final model, a fitness of good control was performed based on Hosmer–Lemeshow criterion (*p*-value = 0.47).

## Data Availability

Data sharing is not applicable to this article.

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
