# Peer review of "Novel Insights for Patients with Multiple Basal Cell Carcinomas and Tumors at High-Risk for Recurrence: Risk Factors, Clinical Morphology, and Dermatoscopy"

_cancers, 2021, doi:10.3390/cancers13133208_

Round 1

Reviewer 1 Report

Dear authors , the article is interesting and innovative, well documeted

Only a minor comments:

-it isn't clear the correlation of the two different objective of the paper , is clear in the methods selection , but better explain the results and the discussion of the two different goal

-table 4-5-6 are too "big" please try to summarized the results in a smaller table 

these suggestions will help the reader to better understand the paper , in this form results too complex to be interpreted

Author Response

We thank the reviewer for the valuable comments.

We agree with reviewer’s revisions.

  • Tables 4-7 are now merged into table 4 and table 5.

-       Our main objectives were the detection of risk factors for multiple BCC in individual patients and the description of clinical and dermatoscopic features of low and high-risk for local recurrence tumors. We strongly support that multiple-BCC patients’ profiling and thorough knowledge of high-risk tumors’ clinico-dermatoscopic morphology could provide physicians with important information towards prevention of this neoplasm.

We accept that the results of the study and the subsequent analysis are extended. However, we had to analyze a large number of variables ranging from behavioral risk factors (e.g. sun exposure) to dermatoscopic criteria. In terms of dermatoscopy, we decided to present them in a very detailed manner in order to be in line with what is commonly accepted for the accuracy of these types of study. Thus, we designed many tables and we had a meticulous description and discussion of our results aiming to give the reader the chance to focus on the most important findings that could be of help for the everyday clinical practice.      

Reviewer 2 Report

Outstanding article with useful data for multiple

BCC management. 

I suggest to add few words on the neoadjuvant

role of new therapies (see “Evolving Role of

Systemic Therapies in Non-melanoma Skin

Cancer” article) for which dermoscopy can help

to identify and measure the size reduction of tumors.no other suggestions 

Author Response

We thank the reviewer for his/her kind comments.

  1. The following phrase has been added in the introduction section:

“Beyond its diagnostic role dermatoscopy can help as a follow-up measure for size reduction of locally advanced BCC under neo-adjuvant systemic treatments” [reference 15 is cited Conforti C, Corneli P, Harwood C, Zalaudek I. Evolving Role of Systemic Therapies in Non-melanoma Skin Cancer. Clin Oncol (R Coll Radiol). 2019 Nov;31(11):759-768. doi: 10.1016/j.clon.2019.08.011. Epub 2019 Sep 12. PMID: 31522944.]

(lines 11-113)

Reviewer 3 Report

44-45. guidelines-approved, non-surgical therapeutic options for specific subtypes of BCC (i.e. superficial)

Do the authors mean “e.g.” instead of “i.e.” as also other subtypes are suitable for non-surgical options.

57- 58. Of note is the absolute correlation of unanimously accepted dermatoscopic criteria with histological subtypes of BCC [15-18].

This strong notion should be founded in high level evidence that are lacking.

70-71. severely immunocompromised patients,.

Need definition.

76-78. Both clinical and dermatoscopic images of the suspicious lesions were captured at the initial medical visit with the use of a Nikon J1 camera and a handheld Dermlite Hybrid II dermatoscope.

State if all excised lesion at the institution was captured before excision or if this is an inclusion criterion.

  1. 1 Any risk factor places the patient in the high-risk category

Mixing, without sub-analyzing risk factors may be a fundamental problem in this study. Hard to follow if location, size, clinical type or histopathologic type are accountable for the findings.

81-82. The cohort was divided into two groups of patients; those with solitary neoplasms and those with more than one tumors (solitary Vs. multiple BCC).

Is this referring to a history of multiple BCC or synchronous? 81-82 This is explained at 114-115 but should be included in M&M

  1. In specific, 24/53 (45.28%) reported occupational sun exposure, 19/53

Consider using one decimal (from 120 and forward).

  1. only 3 patients (3/53, 5.6%).

Remove ”only”.

154-155. Regarding important factors of risk stratification for local recurrence (such as diameter, anatomic site and histopathology) no striking variability was recorded between the two groups of patients.

In patients with e.g. multiple melanomas, the second melanoma is more often thinner than the first, probably due to a screening effect. Do this apply to the second basalioma in this study? How does referral to a tertiary hospital affect patient selection on single basalioma and screening efforts for secondary basalioma?

  1. leaf-like structures at the periphery (whit asterisks), concentric structures (white circle) 228

Check spelling for “whit”.

Author Response

We thank the reviewer for the valuable comments and suggestions.

Please find attached the point-by-point responses:

  1. 44-45. guidelines-approved, non-surgical therapeutic options for specific subtypes of BCC (i.e. superficial)

Do the authors mean “e.g.” instead of “i.e.” as also other subtypes are suitable for non-surgical options.

Response to reviewer’s comment:

"i.e." deleted and replaced by the most accurate "e.g." (line 56)

  1. 57- 58. Of note is the absolute correlation of unanimously accepted dermatoscopic criteria with histological subtypes of BCC [15-18].

This strong notion should be founded in high level evidence that are lacking.

Response to reviewer’s comment:

Reviewer is correct. Current dermatoscopic criteria are widely accepted and efficiently used in everyday clinical practice, despite lacking high level of evidence. The sentence is rephrased more precisely. (lines 71-72)

  1. Severely immunocompromised patients

Need definition

Response to reviewer’s comment:

The explanation for excluded severely immunocopmpromised patients is given respectively.

“(i.e. patients under immunosuppressive treatment for autoimmune diseases or internal malignancies and HIV patients)” is added in the section of exclusion criteria. (lines 85-86)

  1. 76-78. Both clinical and dermatoscopic images of the suspicious lesions were captured at the initial medical visit with the use of a Nikon J1 camera and a handheld Dermlite Hybrid II dermatoscope.

State if all excised lesion at the institution was captured before excision or if this is an inclusion criterion.

Response to reviewer’s comment:

The reviewer’s comment is clarified. The sentence is rephrased accordingly.

“Both clinical and dermatoscopic images of all the suspicious lesions were captured at the initial medical visit before surgical excision with the use of a Nikon J1 camera and a handheld Dermlite Hybrid II dermatoscope.” (lines 92-94)

  1. Any risk factor places the patient in the high-risk category

Mixing, without sub-analyzing risk factors may be a fundamental problem in this study. Hard to follow if location, size, clinical type or histopathologic type are accountable for the findings.

Response to reviewer’s comment:

Reviewer 3 has a strong point with his comment. Without judging and assessing the impact of each criterion individually for high risk BCCs makes this classification system look weak. We agree with reviewer’s statement and we added this in the study limitations. However, this is the currently accepted classification system for BCC since a TNM system is not applicable due to the neoplasm’s extremely low metastatic potential. The fundamental issue could be identified for this system. Our study was based on current evidence and practice for risk stratification for BCCs. In addition, we excluded other criteria such as immunosuppression, tumor’s margins and recurrent neoplasms since we attempted to make a more precise approach to this widely used system, which is also critical for treatment selection. (please check comment in table 1, line 119 and limitations, lines 355-357)

  1. The cohort was divided into two groups of patients; those with solitary neoplasms and those with more than one tumors (solitary Vs. multiple BCC).

Is this referring to a history of multiple BCC or synchronous? 81-82 This is explained at 114-115 but should be included in M&M

Response to reviewer’s comment:

“…more than one synchronous tumors…” is added in the section of materials and methods (line 98)

  1. In specific, 24/53 (45.28%) reported occupational sun exposure, 19/53

Consider using one decimal (from 120 and forward).

 Response to reviewer’s comment:

We have remodeled the percentages presentation with one decimal in the text as well as the tables.

  1. Only 3 patients (3/53, 5.6%).

Remove ”only”.

Response to reviewer’s comment:

The word “only” is deleted (line 170)

  1. Regarding important factors of risk stratification for local recurrence (such as diameter, anatomic site and histopathology) no striking variability was recorded between the two groups of patients.

In patients with e.g. multiple melanomas, the second melanoma is more often thinner than the first, probably due to a screening effect. Do this apply to the second basalioma in this study? How does referral to a tertiary hospital affect patient selection on single basalioma and screening efforts for secondary basalioma?

 Response to reviewer’s comment:

We thank the reviewer for the comment and we agree. The visit-seeking attitude of patients with personal history of BCC and the surveillance plan after diagnosis of an initial tumor might pose a bias. We added this in the study limitations. (please check comment in lines 170 and 357-359)

  1. leaf-like structures at the periphery (whit asterisks), concentric structures (white circle) 228

Check spelling for “whit”.

Response to reviewer’s comment:

Typing error that is corrected “white” (line 240)
